# Video-Text Retrieval by Supervised Sparse Multi-Grained Learning

**Yimu Wang**
University of Waterloo
yimu.wang@uwaterloo.ca

**Peng Shi**
University of Waterloo
peng.shi@uwaterloo.ca

## Abstract

While recent progress in video-text retrieval has been advanced by the exploration of better representation learning, in this paper, we present a novel multi-grained sparse learning framework, S3MA, to learn an aligned sparse space shared between the video and the text for video-text retrieval. The shared sparse space is initialized with a finite number of sparse concepts, each of which refers to a number of words. With the text data at hand, we learn and update the shared sparse space in a supervised manner using the proposed similarity and alignment losses. Moreover, to enable multi-grained alignment, we incorporate frame representations for better modeling the video modality and calculating fine-grained and coarse-grained similarities. Benefiting from the learned shared sparse space and multi-grained similarities, extensive experiments on several video-text retrieval benchmarks demonstrate the superiority of S3MA over existing methods. Our code is available at link.

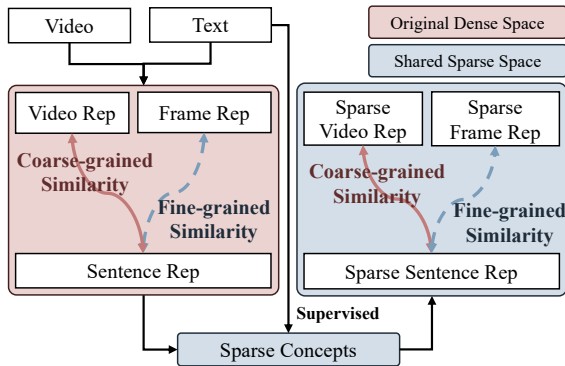

Figure 1: Our proposed *supervised shared sparse multi-grained alignment* framework for video-text retrieval maps sentence, video, and frame representations to a shared sparse space to obtain sparse sentence, video, and frame representations. Then, it calculates *coarse- and fine-grained* similarities to fully explore the power of the sparse space, which is learned in a *supervised* fashion. "Original Dense Space" represents the space containing the representations generated from modality-dependent encoders. "Shared Sparse Space" represents the space containing the sparse concepts shared across two modalities. "Rep" refers to representation.

## 1 Introduction

As a fundamental task in visual-language understanding (Wang et al., 2020b; Xu et al., 2021b; Park et al., 2022a; Miyawaki et al., 2022; Fang et al., 2023a,b; Kim et al., 2023; Jian and Wang, 2023), video-text retrieval (VTR) (Luo et al., 2022; Gao et al., 2021b; Ma et al., 2022a; Liu et al., 2022a; Zhao et al., 2022; Gorti et al., 2022; Fang et al., 2022) has attracted interest from academia and industry. Although recent years have witnessed the rapid development of VTR with the support from powerful pretraining models (Luo et al., 2022; Gao et al., 2021b; Ma et al., 2022a; Liu et al., 2022a), improved retrieval methods (Bertasius et al., 2021; Dong et al., 2019; Jin et al., 2021), and video-language datasets construction (Xu et al., 2016), it remains challenging to precisely match video and language due to the raw data being in heterogeneous spaces with significant differences.

Current VTR research (Luo et al., 2022; Ma et al., 2022a; Liu et al., 2022b) mainly aims to learn a joint feature space across modalities and then compares representations in this space. However, with the huge discrepancy between different modalities and the design of modality-independent encoders, it is challenging to directly compare and calculate the similarities between representations of different modalities generated from different encoders (Liang et al., 2022). To alleviate the mismatch caused by heterogeneous encoders and data formats, Liu et al. (2022a); Cao et al. (2022) proposed to align different modalities in a common space without supervision from text or video. However, because of the unsupervised design, the shared spaces are either randomly initialized or updated in an unsupervised fashion, which blocks the power of that aligned space. We argue that learning

a shared aligned space with supervision is a promising way to improve video-text retrieval. Borrowing from text retrieval (Karpukhin et al., 2020; Zhao et al., 2021; Gao et al., 2021a), we represent the aligned space and the space containing representations generated by modality-dependent encoders as sparse and dense spaces, respectively, as the aligned space typically carries specific semantics.

In this work, we propose a *Supervised Shared Sparse Multi-grained Alignment framework* for VTR, namely S3MA, in which the aligned sparse space is updated under the supervision of the video-text data at hand. Specifically, we initialize a finite number of sparse concepts by clustering a large number of basic concepts (words) to form the fine-grained aligned sparse space. In return, each sparse concept is composed of several words, which improves the interpretability of our model. Then, we match the sparse text and video representations effectively by projecting the video representation generated by the video encoder to this fine-grained sparse space. The sparse sentence (text) representations can be obtained by looking up the sparse concepts. To obtain sparse video representations, we first calculate the cosine similarity between the video representations and the sparse concepts. Next, by summing up all the sparse concepts with the weight of the cosine similarity between video representation and sparse concepts, we obtain the sparse video representations. Furthermore, to better match these two sparse representations, we design two loss functions to update sparse concepts, pushing the sparse representations of text and video as close as possible in the shared sparse space. This shared sparse space design not only improves the performance on VTR, but also allows us to interpret what the models have learned. The sparse aligned space, as shown in Figure 5, enables the model to accurately capture the key concepts, resulting in improved alignment within the sparse space.

Recently, Ma et al. (2022a) demonstrated that incorporating fine-grained video representations (such as frame or segment representations) with high-level video features can further improve retrieval performance. Inspired by their work, we further project *frame* representations into our designed aligned sparse space. Compared to high-level video representations, frame representations can be mapped to more detailed concepts, which enriches the overall video representations. In this way, we have fine-grained (frame) and coarse-grained (video and sentence) representations from the sparse space and the dense space, enabling us to calculate multi-space multi-grained similarity for exploring the potential of supervised sparse space.

Finally, to evaluate the effectiveness of our proposed S3MA, we conducted experiments on three video-text benchmarks (Chen and Dolan, 2011; Fabian Caba Heilbron and Niebles, 2015; Xu et al., 2016). Benefiting from multi-grained and multi-space similarity, our proposed S3MA outperforms previous methods on all the benchmarks without requiring any additional data during training.

In summary, our contributions are as follows[1]:

- We propose the shared sparse space to alleviate the problem of mismatched representations from different modalities, which arises from the raw data being in heterogeneous spaces and the heterogeneous design of modality-dependent encoders.

- Our proposed S3MA achieves SOTA performance on several metrics across three VTR benchmarks.

- Detailed analysis reveals the importance of shared sparse space and multi-grained similarity. Besides, we demonstrate that the design of shared sparse space and multi-grained similarity significantly impacts retrieval performance.

## 2 Related Works

Video-Text Retrieval (VTR), which involves cross-modal alignment and abstract understanding of temporal images (videos), has been a popular and fundamental task of language-grounding problems (Wang et al., 2020a,c, 2021; Yu et al., 2023). Most existing conventional video-text retrieval frameworks (Yu et al., 2017; Dong et al., 2019; Zhu and Yang, 2020; Miech et al., 2020; Gabeur et al., 2020; Dzabraev et al., 2021; Croitoru et al., 2021) focus on learning powerful representations for video and text and extracting separated representations. Inspired by the success of self-supervised pretraining methods (Devlin et al., 2019; Radford et al., 2019; Brown et al., 2020) and vision-language pretraining (Li et al., 2020b; Gan et al., 2020; Singh et al., 2022) on large-scale unlabeled cross-modal data, recent works (Lei et al., 2021; Cheng et al., 2021; Gao et al., 2021b; Ma et al.,

---

[1]The code is released at link.

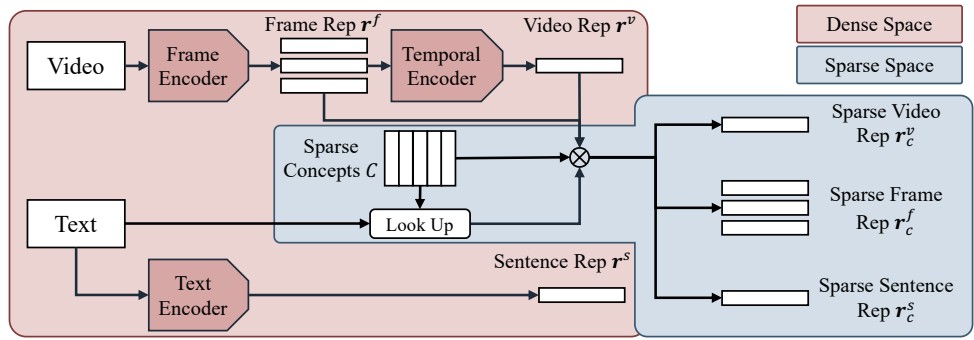

Figure 2: The illustration of representation generation in our proposed *Supervised Shared Sparse Multi-grained Alignment framework*, namely S3MA. Specifically, for multi-space alignment, we employ a shared sparse space which is consisted of a number of sparse concepts. The shared sparse space is updated in a supervised manner during the training procedure, leading to the construction of a fine-grained sparse space. "⊗" refers to the calculation in Eqs. (1), (2), and (3).

2022a; Park et al., 2022a; Wang et al., 2022b,c; Zhao et al., 2022; Gorti et al., 2022) have attempted to pretrain or fine-tune video-text retrieval models in an end-to-end manner. Frozen in time (Bain et al., 2021) uses end-to-end training on both image-text and video-text pairs data by uniformly sampling video frames. CLIP4Clip (Luo et al., 2022) finetunes models and investigates three similarity calculation approaches for video-sentence contrastive learning on CLIP (Radford et al., 2021). Later, to enable unsupervised sparse learning in VTR, DiscretCodebook (Liu et al., 2022a) aligns modalities in a shared space filled with concepts, which are randomly initialized and unsupervisedly updated, while VCM (Cao et al., 2022) constructs a sparse space with unsupervisedly clustered visual concepts. At the same time, OA-Trans (Wang et al., 2022a) and TABLE (Chen et al., 2023) both employ a small number of semantic tags as the input to the text encoder to improve alignment between modalities.

However, due to the unsupervised design, concepts in DiscretCodebook and VCM are either randomly initialized or updated unsupervisedly, which limits the potential of aligned sparse space. On the other hand, OA-Trans and TABLE only employ a limited number of concepts to serve as the input of the text encoder to encourage alignment. Meanwhile, these methods only perform the *coarse-grained* video-text similarity, lacking the fine-grained contrast between different modalities. In comparison, our proposed S3MA learn the aligned sparse space containing a large number of words in a *supervised* manner, under the supervision of text, and calculate frame-sentence similarity

for *multi-space multi-grained* alignment.

## 3 Methods

In this section, we introduce our proposed framework for video-text retrieval, which aligns language and video in a shared sparse space. Typically, in video-text retrieval, we have a set of examples $\{(\mathbf{v}_i, \mathbf{t}_i)\}_{i \in [N]}$, where $N$ is the number of examples that are of video and language.

### 3.1 General Video-Text Retrieval Paradigm

In this part, we present a general video-text retrieval framework widely used by previous methods (Luo et al., 2022; Liu et al., 2022a). With this paradigm, we can obtain three representations for different modalities from the dense space, *i.e.*, frame representation $\mathbf{r}^f$, video representation $\mathbf{r}^v$, and sentence representation $\mathbf{r}^s$ by modality-dependent encoders.

**Frame and video representations:** Given a video $\mathbf{v}$, several video frames are first sampled as the inputs of the frame encoder to obtain the frame features $\mathbf{r}^f \in \mathbb{R}^{n_{\mathbf{t}_{frame}} \times d}$, where $n_{frame}$ is the number of frames and $d$ is the dimension of features. As the frame representations $\mathbf{r}^f$ are extracted through sampling, to explore the temporal correlation among different frames, we employ a temporal encoder to aggregate frame representations. With the temporal encoder and the frame representations $\mathbf{r}^f$, we obtain the video representations $\mathbf{r}^v \in \mathbb{R}^{1 \times d}$.

**Sentence representation:** Given a sentence $\mathbf{t}$, we use a text encoder to obtain the text representation $\mathbf{r}^s \in \mathbb{R}^{1 \times d}$.

## 3.2 Fine-Grained Aligned Sparse Space

The key to the video-text retrieval task is to precisely align representations from different modalities. However, due to the heterogeneous encoder architectures and data formats of different modalities, it is difficult to align directly (Liang et al., 2022). Therefore, instead of directly enforcing the representations to be aligned, we propose aligning them in an aligned sparse constructed by $n_c$ sparse concepts $C \in \mathbb{R}^{n_c \times d}$. Each sparse concept $\mathbf{c}$ represents several basic concepts (words). Moreover, to supervise the updates of sparse concepts, we utilize the human-annotated knowledge at hand, *i.e.*, text annotations in the paired video-text data.

**Initialization**. First, we map all the words into embeddings by the embedding layer $f_{emb}$ of the text encoder. But as the number of words is relatively large (for example, in Clip (Radford et al., 2021), the number of sub-words is approximately 30k), we cluster embeddings into $n_c$ clusters using KNN (Gianfelici, 2008) to form the sparse concepts $C$ and represent all the words by their cluster's centers $\mathbf{c}$. Consequently, each sparse concept $\mathbf{c}$ represents a bunch of words that are similar on the embedding space, enabling fine-grained alignment. The mapping from words to sparse concepts is denoted by $h_{w2c} \in [n_{words}] \to \{0,1\}^{n_c \times 1}$. Now, $n_c$ sparse concepts have been initialized.

**Obtaining the sparse sentence representation**. For text, as the caption is at hand, we can directly tokenize the sentences into words and look up the corresponding sparse concepts in $C$. The sparse sentence representation $\mathbf{r}_c^s \in \mathbb{R}^{1 \times d}$ is obtained by averaging all the representations of concepts that are fetched with the surface form of the sentence, as follows,

$$\mathbf{r}_c^s = sim^{\mathbf{t}\top} C / |\mathbf{t}|, \tag{1}$$

where $|\mathbf{t}|$ is the number of words in $\mathbf{t}$ and $sim^{\mathbf{t}} = \sum_{w \in \mathbf{t}} h_{w2c}(w)$ is a vector with the length of $n_c$.

**Obtaining the sparse video representation**. We first calculate the cosine similarity $sim^{\mathbf{v}} \in \mathbb{R}^{1 \times n_c}$ between the video representations and sparse concepts $C$ as $sim_j^{\mathbf{v}} = \cos(\mathbf{r}^v, \mathbf{c}_j), \forall j \in [n_c]$, where $sim_j^{\mathbf{v}}$ is the $j$-th element of $sim^{\mathbf{v}}$ and $\cos(\cdot, \cdot)$ is the cosine similarity. Next, sparse video representations are obtained by weighted summing the sparse concepts as,

$$\mathbf{r}_c^v = sim^{\mathbf{v}} C / \|sim^{\mathbf{v}}\|_1. \tag{2}$$

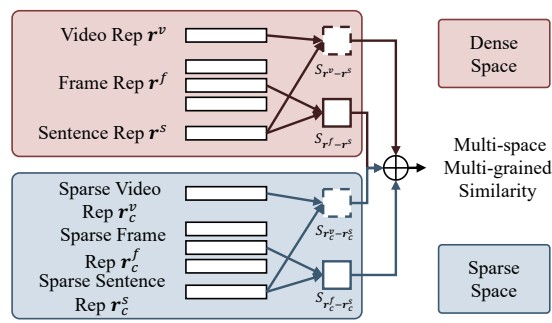

Figure 3: The illustration of similarity calculation. To enable multi-space multi-grained alignment, we calculate fine-grained (frame-sentence) and coarse-grained (video-sentence) similarity. Our preliminary experiments showed that the text encoder has a good ability to capture semantics, so we only use sentence representations for the text modality.

**Obtaining the sparse frame representation**. Similarly, the cosine similarity $sim^f \in \mathbb{R}^{n_{frame} \times n_c}$ between the frame representations and sparse concepts is calculated as $sim_{i,j}^f = \cos(\mathbf{r}_i^f, \mathbf{c}_j), \forall i \in [n_{frame}], \forall j \in [n_c]$, where $sim_{i,j}^f$ is the $(i,j)$-th element of $sim^f$ and $\mathbf{r}_i^f$ is the $i$-th row of $\mathbf{r}^f$. Next, sparse frame representations are obtained as,

$$\mathbf{r}_c^f = \sum_{i \in [n_{frame}]} sim_i^f C / \|sim_i^f\|_1. \tag{3}$$

Finally, we have the sparse frame, video, and sentence representations $\mathbf{r}_c^f \in \mathbb{R}^{n_{frame} \times d}, \mathbf{r}_c^v \in \mathbb{R}^{1 \times d}, \mathbf{r}_c^s \in \mathbb{R}^{1 \times d}$ with the frame and video sparse space similarity $sim^f \in \mathbb{R}^{n_{frame} \times n_c}$ and $sim^{\mathbf{v}} \in \mathbb{R}^{n_c}$ along with the sentence sparse space similarity (supervision) $sim^{\mathbf{t}}$.

## 3.3 Multi-Space Multi-Grained Similarity

In this part, we will demonstrate our method for calculating the similarities between data from two different modalities, as shown in Figure 3, including the similarities in the dense space and in shared sparse space, inspired by Ma et al. (2022a). We can now compute multi-space (sparse and dense spaces) multi-grained (fine-grained and coarse-grained) similarity for precise alignment.

### 3.3.1 Dense Space Similarity

**Video-Sentence similarity** $S_{\mathbf{r}^v - \mathbf{r}^s}$. To obtain a fine-grained similarity, we use a learnable matrix $A_{\mathbf{r}^v - \mathbf{r}^s} \in \mathbb{R}^{d \times d}$ to focus on the discriminative features of video and sentence representations as,

$$S_{\mathbf{r}^v - \mathbf{r}^s} = \mathbf{r}^v A_{\mathbf{r}^v - \mathbf{r}^s} \mathbf{r}^{s\top}.$$

**Frame-Sentence similarity** $S_{\mathbf{r}^f - \mathbf{r}^s}$. To obtain a fine-grained similarity, we first calculate an *instance-aware weight* using the softmax function applied to the dot product of $\mathbf{r}^s \mathbf{r}^{f\top}$, and then use a learnable matrix $A_{\mathbf{r}^f - \mathbf{r}^s} \in \mathbb{R}^{n_{frame} \times n_{frame}}$ to focus on discriminative frames. In this way, the similarity is calculated as,

$$S_{\mathbf{r}^f - \mathbf{r}^s} = \text{softmax}(\mathbf{r}^s \mathbf{r}^{f\top}) A_{\mathbf{r}^f - \mathbf{r}^s} \mathbf{r}^f \mathbf{r}^{s\top} .$$

### 3.3.2 Sparse Space Similarity

**Video-Sentence shared sparse space similarity** $S_{\mathbf{r}^v_c - \mathbf{r}^s_c}$. Similarly, to obtain a fine-grained similarity on the shared sparse space, we use a learnable matrix $A_{\mathbf{r}^v_c - \mathbf{r}^s_c} \in \mathbb{R}^{d \times d}$ to focus on the discriminative features of sparse video and sentence representations. Now, the similarity is calculated as,

$$S_{\mathbf{r}^v_c - \mathbf{r}^s_c} = \mathbf{r}^v_c A_{\mathbf{r}^v_c - \mathbf{r}^s_c} \mathbf{r}^{s\top}_c .$$

**Frame-Sentence shared sparse space similarity** $S_{\mathbf{r}^f_c - \mathbf{r}^s_c}$. With *instance-aware weights* $\text{softmax}(\mathbf{r}^s_c \mathbf{r}^{f\top}_c)$ and a learnable matrix $A_{\mathbf{r}^f_c - \mathbf{r}^s_c} \in \mathbb{R}^{n_{frame} \times n_{frame}}$, we get the similarity between the sparse frame and sentence representations as,

$$S_{\mathbf{r}^f_c - \mathbf{r}^s_c} = \text{softmax}(\mathbf{r}^s_c \mathbf{r}^{f\top}_c) A_{\mathbf{r}^f_c - \mathbf{r}^s_c} \mathbf{r}^f_c \mathbf{r}^{s\top}_c .$$

### 3.3.3 Overall Similarity

The overall video-text similarity is defined as,

$$S = \frac{S_{\mathbf{r}^f - \mathbf{r}^s} + S_{\mathbf{r}^v - \mathbf{r}^s} + S_{\mathbf{r}^f_c - \mathbf{r}^s_c} + S_{\mathbf{r}^v_c - \mathbf{r}^s_c}}{4} .$$

### 3.4 Objective

The objective consists of three different losses. The first component is contrastive loss. Following Clip4Clip (Luo et al., 2022), we employ the symmetric InfoNCE loss over the similarity matrix to optimize the retrieval model as,

$$
\begin{aligned}
\ell_{sim} =& \ell_{v2t} + \ell_{t2v} \\
=& -\frac{1}{N} \sum_{i \in [N]} \log \frac{\exp(S_{i,i})}{\sum_{j \in [N]} \exp(S_{i,j})} \\
& -\frac{1}{N} \sum_{i \in [N]} \log \frac{\exp(S_{i,i})}{\sum_{j \in [N]} \exp(S_{j,i})} ,
\end{aligned}
$$

where $S_{i,j}$ is similarity between $i$-th video and $j$-th text and $N$ is the number of paired data.

The second loss we minimize is the alignment loss, which matches the sparse frame and video

representations ($\mathbf{r}^f_c$ and $\mathbf{r}^v_c$) with the sparse sentence representations $\mathbf{r}^s_c$ in the $\ell_2$ distance, as,

$$
\begin{aligned}
\ell_{align} =& \frac{1}{N} \sum_{i \in [N]} \left( \| \mathbf{r}^v_c - \mathbf{r}^s_c \|_2 \right. \\
& \left. + \left\| \frac{\mathbf{1} \mathbf{r}^f_c}{n_{frame}} - \mathbf{r}^s_c \right\|_2 \right) ,
\end{aligned}
$$

where $\mathbf{1}$ is the vector only containing 1.

In addition, to match the frame and video representations with the corresponding sparse concepts, we minimize the sparse similarity loss as,

$$
\begin{aligned}
\ell_{sparse} =& \frac{1}{N} \sum_{i \in [N]} \left( \| sim^{\mathbf{v}} - sim^{\mathbf{t}} \|_2 \right. \\
& \left. + \left\| \frac{\mathbf{1} sim^f}{n_{frame}} - sim^{\mathbf{t}} \right\|_2 \right) ,
\end{aligned}
$$

The overall objective is the linear combination of the above three losses as,

$$\ell = \ell_{sim} + \alpha \ell_{align} + \beta \ell_{sparse} ,$$

where $\alpha$ and $\beta$ are hyperparameters controlling the trade-off between three losses. We set $\alpha = 0.02$ and $\beta = 0.01$ for all the experiments.

## 4 Experiments

### 4.1 Datasets and Baselines

To show the empirical efficiency of our S3MA, we train it on MSR-VTT (Xu et al., 2016), MSVD (Chen and Dolan, 2011), and ActivityNet (Fabian Caba Heilbron and Niebles, 2015). We compare with VLM (Xu et al., 2021a), HERO (Li et al., 2020a), VideoCLIP (Xu et al., 2021b), EvO (Shvetsova et al., 2022), OA-Trans (Wang et al., 2022a), RaP (Wu et al., 2022), LiteVL (Chen et al., 2022), NCL (Park et al., 2022b), TABLE (Chen et al., 2023), VOP (Huang et al., 2023), Clip4Clip (Luo et al., 2022), X-CLIP (Ma et al., 2022a), DiscreteCodebook (Liu et al., 2022a), TS2-Net (Liu et al., 2022b), VCM (Cao et al., 2022), HiSE (Wang et al., 2022b), Align&Tell (Wang et al., 2022c), Center-CLIP (Zhao et al., 2022), and X-Pool (Gorti et al., 2022). Implementation details and evaluation protocols are deferred to the Appendix.

### 4.2 Quantitative Results

**MSR-VTT.** As shown in Table 1, S3MA achieves the best R@1 on the text-to-video retrieval results

| Methods | Venue | Text-to-Video Retrieval | | | | | Video-to-Text Retrieval | | | | |
|---|---|---|---|---|---|---|---|---|---|---|---|
| | | R@1↑ | R@5↑ | R@10↑ | MdR↓ | MnR↓ | R@1↑ | R@5↑ | R@10↑ | MdR↓ | MnR↓ |
| VLM | ACL'21 | 28.1 | 55.5 | 67.4 | 4.0 | - | - | - | - | - | - |
| HERO | EMNLP'21 | 16.8 | 43.3 | 57.7 | - | - | - | - | - | - | - |
| VideoCLIP | EMNLP'21 | 30.9 | 55.4 | 66.8 | - | - | - | - | - | - | - |
| EvO | CVPR'22 | 23.7 | 52.1 | 63.7 | 4.0 | - | - | - | - | - | - |
| OA-Trans | CVPR'22 | 35.8 | 63.4 | 76.5 | 3.0 | - | - | - | - | - | - |
| RaP | EMNLP'22 | 40.9 | 67.2 | 76.9 | 2.0 | - | - | - | - | - | - |
| *BLIP-based* | | | | | | | | | | | |
| LiteVL-S | EMNLP'22 | 46.7 | 71.8 | 81.7 | 2.0 | - | - | - | - | - | - |
| *ViT-B/32-based* | | | | | | | | | | | |
| Align&Tell | TMM | 45.2 | 73.0 | 82.9 | 2.0 | - | 43.4 | 70.9 | 81.8 | 2.0 | - |
| X-Pool | CVPR'22 | 46.9 | 72.8 | 82.2 | 2.0 | 14.3 | - | - | - | - | - |
| CenterCLIP | SIGIR'22 | 44.2 | 71.6 | 82.1 | 2.0 | 15.1 | 42.8 | 71.7 | 82.2 | 2.0 | 10.9 |
| TS2-Net | ECCV'22 | 47.0 | 74.5 | 83.8 | 2.0 | 13.0 | 45.3 | 74.1 | 83.7 | 2.0 | 9.2 |
| X-CLIP | ACM MM'22 | 46.1 | 74.3 | 83.1 | 2.0 | 13.2 | 46.8 | 73.3 | 84.0 | 2.0 | 9.1 |
| NCL | EMNLP'22 | 43.9 | 71.2 | 81.5 | 2.0 | 15.5 | 44.9 | 71.8 | 80.7 | 2.0 | 12.8 |
| TABLE | AAAI'23 | 47.1 | 74.3 | 82.9 | 2.0 | 13.4 | 47.2 | 74.2 | 84.2 | 2.0 | 11.0 |
| VOP | CVPR'23 | 44.6 | 69.9 | 80.3 | 2.0 | 16.3 | 44.5 | 70.7 | 80.6 | 2.0 | 11.5 |
| CLIP4Clip | NC | 44.5 | 71.4 | 81.6 | 2.0 | 15.3 | - | - | - | - | - |
| DiscreteCodebook | ACL'22 | 43.4 | 72.3 | 81.2 | - | 14.8 | 42.5 | 71.2 | 81.1 | - | 12.0 |
| VCM | AAAI'22 | 43.8 | 71.0 | - | 2.0 | 14.3 | 45.1 | 72.3 | 82.3 | 2.0 | 10.7 |
| S3MA | | 49.1 | 73.9 | 82.8 | 2.0 | 13.5 | 46.9 | 73.8 | 82.1 | 2.0 | 9.3 |
| S3MA† | | **51.7** | **75.9** | **85.4** | 1.0 | **11.1** | 51.6 | **76.8** | **85.0** | 1.0 | **8.4** |
| *ViT-B/16-based* | | | | | | | | | | | |
| Align&Tell | TMM | 47.4 | 74.3 | 84.1 | 2.0 | - | 45.3 | 73.5 | 83.7 | 2.0 | - |
| CenterCLIP | SIGIR'22 | 48.4 | 73.8 | 82.0 | 2.0 | 13.8 | 47.7 | 75.0 | 83.3 | 2.0 | 10.2 |
| HiSE | ACM MM'22 | 45.0 | 72.7 | 81.3 | 2.0 | - | 46.6 | 73.3 | 82.3 | 2.0 | - |
| TS2-Net | ECCV'22 | 49.4 | 75.6 | 85.3 | 2.0 | 13.5 | 46.6 | 75.9 | 84.9 | 2.0 | 8.9 |
| CLIP4Clip | NC | 45.8* | 74.3* | 84.1* | 2.0* | - | 43.2* | 71.3* | 82.0* | 2.0* | - |
| S3MA | | 49.8 | 75.1 | 83.9 | 2.0 | 12.2 | 47.3 | 76.0 | 84.3 | 2.0 | 8.9 |
| S3MA† | | **53.1** | **78.2** | **86.2** | 1.0 | **10.5** | **52.7** | **79.2** | **86.3** | 1.0 | **8.2** |

Table 1: Video-Text retrieval results on MSR-VTT. * represents data copied from Align&Tell. The best results are marked in **bold**. The second best results are underlined. "NC" refers to Neurocomputing. † refers to the results with the inverted softmax.

| Methods | Venue | Text-to-Video Retrieval | | | |
|---|---|---|---|---|---|
| | | R@1↑ | R@5↑ | R@10↑ | MnR↓ |
| *MSVD* | | | | | |
| X-CLIP | ACM MM'22 | 47.1 | 77.8 | - | 9.5 |
| HiSE | ACM MM'22 | 45.9 | 76.2 | 84.6 | - |
| X-Pool | CVPR'22 | 47.2 | 77.4 | **86.0** | 9.3 |
| CLIP4Clip | NC | 45.2 | 75.5 | 84.3 | 10.3 |
| S3MA | | **47.3** | 78.8 | 85.7 | 9.3 |
| *ActivityNet* | | | | | |
| Align&Tell | TMM | 42.6 | 73.8 | - | - |
| X-CLIP | ACM MM'22 | 44.3 | 74.1 | - | 7.9 |
| TS2-Net | ECCV'22 | 41.0 | 73.6 | 84.5 | 8.4 |
| CLIP4Clip | NC | 40.5 | 72.4 | - | 7.5 |
| VCM | AAAI'22 | 40.8 | 72.8 | - | 7.3 |
| S3MA | | **45.0** | **75.5** | **85.7** | **6.3** |

Table 2: Text-Video retrieval results on MSVD and ActivityNet. The best results are marked in **bold**. The second best results are underlined.

using ViT-B/32 and ViT-B/16, outperforming the second-best method by 2.1 and 0.4, respectively.

The performance of S3MA on the video-to-text retrieval task is also comparable with previous methods, achieving the best and second-best results on R@1 and R@5 using ViT-B/32. Moreover, we notice that only 1 previous method using ViT-B/16 outperforms S3MA with ViT-B/32 on the text-to-video retrieval, demonstrating the effectiveness of S3MA. Compared to DiscreteCodebook (Liu et al., 2022a), which aligns modalities in an unsupervised manner, S3MA outperforms DiscreteCodebook on every metric. Meanwhile, S3MA also outperforms VCM (Cao et al., 2022), which constructs an aligned space with unsupervisedly clustered visual concepts, demonstrating the importance of supervising alignment in the sparse space. This suggests that aligning modalities with fine-grained supervision is a promising approach to improving video-to-text retrieval performance.

**MSVD and ActivityNet.** The results on MSVD

| | Text-to-Video Retrieval | | | | | Video-to-Text Retrieval | | | | |
|---|---|---|---|---|---|---|---|---|---|---|
| | R@1↑ | R@5↑ | R@10↑ | MdR↓ | MnR↓ | R@1↑ | R@5↑ | R@10↑ | MdR↓ | MnR↓ |
| S3MA (ViT-B/32) w. SE | 47.3 | 73.5 | 82.0 | 2.0 | 13.3 | 45.6 | 73.4 | **82.4** | 2.0 | **9.1** |
| S3MA (ViT-B/32) w. Emb | **49.1** | **73.9** | **82.8** | 2.0 | **13.5** | **46.9** | **73.8** | 82.1 | 2.0 | 9.3 |

Table 3: Comparing the power of different sparse spaces on MSR-VTT. "Emb" and "SE" refers to the embedding space and semantic embedding space.

| | Text-to-Video Retrieval | | | | | Video-to-Text Retrieval | | | | |
|---|---|---|---|---|---|---|---|---|---|---|
| | R@1↑ | R@5↑ | R@10↑ | MdR↓ | MnR↓ | R@1↑ | R@5↑ | R@10↑ | MdR↓ | MnR↓ |
| S3MA (ViT-B/32) w/o clustering | 48.7 | **74.4** | **83.0** | 2.0 | **13.4** | 46.7 | 73.3 | **82.6** | 2.0 | **9.2** |
| S3MA (ViT-B/32) | **49.1** | 73.9 | 82.8 | 2.0 | 13.5 | **46.9** | **73.8** | 82.1 | 2.0 | 9.3 |

Table 4: Ablation study on the effect of clustering when constructing the shared sparse space.

| Size | Text-to-Video Retrieval | | | Video-to-Text Retrieval | | |
|---|---|---|---|---|---|---|
| | R@1 | R@5 | MnR | R@1 | R@5 | MnR |
| 512 | 48.7 | 73.0 | **12.9** | 46.4 | 72.8 | **9.0** |
| 1024 | **49.1** | **73.9** | 13.5 | 46.9 | **73.8** | 9.3 |
| 2048 | 48.3 | **73.9** | 13.5 | **47.0** | 72.7 | 9.1 |
| 4096 | 47.6 | 73.6 | 13.6 | 46.8 | 73.4 | 9.3 |
| DC (1024) | 43.4 | 72.3 | 14.8 | 42.5 | 71.2 | 12.0 |
| VCM | 43.8 | 71.0 | 14.3 | 45.1 | 72.3 | 10.7 |

Table 5: Retrieval performance with different sizes of sparse space on the MSR-VTT dataset using S3MA with ViT/B-32. "DC" represents DiscreteCodebook (Liu et al., 2022a), which also aligns modalities in a sparse space whose size is 1024 with the base model of ViT/B-32. The best results are marked in **bold**. The second best results are underlined.

and ActicityNet are shown in Table 2. S3MA achieves the best R@1 on text-to-video retrieval on two datasets compared to the previous methods. Besides, with the shared sparse space and multi-grained alignment, S3MA also has the lowest MnR.

## 4.3 Ablation Studies

In this part, we present a series of ablation experiments on MSR-VTT to demonstrate the effectiveness of different components of S3MA. The evaluation of two proposed losses, similarity calculation, and the importance of word-level features are deferred to the Appendix.

### 4.3.1 Efficiency of Sparse Space

**The choice of different initialization of sparse spaces**. To choose the best initialization method for the sparse space, we conduct experiments using two different initializations, *i.e.*, the embedding and semantic embedding spaces, as shown in Table 3. The embedding space is the one we use in S3MA,

while the semantic embedding space, is initialized by outputs of the last layer in the text encoder, with input consisting of a word and two [SEP] tokens. By replacing the embedding initialization with the semantic embedding, the retrieval performance of S3MA decreases, proving the superiority of embedding space over the semantic embedding space.

**Size of sparse space.** Another important factor to consider is the size of the sparse space. When we have unlimited data to train models, a large sparse space is ideal. However, when the data is limited, a large sparse space can lead to sparse gradients, resulting in most of the concepts not being able to be updated, while a small sparse space will restrict the retrieval ability as it becomes more challenging to distinguish between numerous data points. The results of these experiments can be found in Table 5. We see that halving and doubling the size of the sparse space slightly decreases performance.

**Impact of clustering**. As S3MA clusters all the embeddings to initialize concept clusters, it is uncertain whether clustering will hinder the power of the shared sparse space. Clustering can be useful to extract high-level abstract concepts and reduce noise. However, it may also lead to a loss of information, which is important for fine-grained alignment. Specifically, we compare the performance of S3MA to that of a modified version, S3MA w/o clustering concepts, which directly uses over 30k basic concepts to form the shared sparse space. Quantitative results can be found in Table 4. The results show that without clustering, R@5, R@10, and MnR on text-to-video retrieval and R@10 and MnR on video-to-text retrieval are improved. On one hand, similar basic concepts can be better separated, which leads to more precise alignment. On the other hand, that may lead to

| Dense Space | | Sparse Space | | Text-to-Video Retrieval | | | | | Video-to-Text Retrieval | | | | |
| S-V | S-F | S-V | S-F | R@1↑ | R@5↑ | R@10↑ | MdR↓ | MnR↓ | R@1↑ | R@5↑ | R@10↑ | MdR↓ | MnR↓ |
|---|---|---|---|---|---|---|---|---|---|---|---|---|---|
| ✓ | | | | 42.8 | 72.0 | 82.3 | 2.0 | 15.0 | 41.9 | 71.1 | 81.5 | 2.0 | 11.1 |
| ✓ | | ✓ | | 43.3 | 70.5 | 81.4 | 2.0 | 15.6 | 42.5 | 71.0 | 80.9 | 2.0 | 11.9 |
| | ✓ | | | 44.4 | 71.8 | 81.8 | 2.0 | 14.5 | 44.1 | 71.8 | 81.7 | 2.0 | 10.4 |
| | ✓ | | ✓ | 44.8 | 72.1 | 81.7 | 2.0 | 15.9 | 41.7 | 70.2 | 79.6 | 2.0 | 10.8 |
| ✓ | | | ✓ | 42.9 | 72.3 | 81.6 | 2.0 | 15.2 | 42.0 | 70.9 | 81.1 | 2.0 | 11.0 |
| | ✓ | ✓ | | 43.8 | 72.1 | 82.3 | 2.0 | 14.7 | 41.5 | 70.6 | 80.3 | 2.0 | 9.8 |
| ✓ | ✓ | | | 44.0 | 71.3 | 80.9 | 2.0 | 14.8 | 43.6 | 69.5 | 80.1 | 2.0 | 10.4 |
| ✓ | ✓ | ✓ | | 47.4 | 73.3 | 82.4 | 2.0 | **12.9** | 46.4 | 73.0 | **82.2** | 2.0 | **8.9** |
| ✓ | ✓ | | ✓ | 47.4 | 73.6 | 82.5 | 2.0 | 13.2 | **47.3** | 72.3 | 81.7 | 2.0 | **8.9** |
| ✓ | ✓ | ✓ | ✓ | **49.1** | **73.9** | **82.8** | 2.0 | 13.5 | 46.9 | **73.8** | 82.1 | 2.0 | 9.3 |

Table 6: Retrieval performance with different similarities on MSR-VTT using S3MA with the base model of ViT-B/32. "S-V" and "S-F" represent Sentence-Video (coarse-grained) and Sentence-Frame (fine-grained) similarities.

| Base Model | TE | Text-to-Video | | | Video-to-Text | | |
|---|---|---|---|---|---|---|---|
| | | R@1 | R@5 | MnR | R@1 | R@5 | MnR |
| ViT-B/32 | | 47.0 | **73.9** | 14.5 | 45.7 | 72.3 | 9.6 |
| | ✓ | **49.1** | **73.9** | 13.5 | **46.9** | **73.8** | **9.3** |
| ViT-B/16 | | 47.3 | 74.9 | 12.8 | 46.1 | 75.1 | 9.5 |
| | ✓ | **49.8** | **75.1** | **12.2** | **47.3** | **76.0** | **8.9** |

Table 7: Retrieval performance with or without the temporal encoder ("TE") and with different base models.

sparse gradients, resulting in some concepts not being fully updated while others are over-updated. This might cause some concepts to be under or over-represented, which might negatively impact the performance (Radovanovic et al., 2010). Therefore, it's important to find the balance in clustering to achieve the best performance.

### 4.3.2 Efficiency of Multi-Grained Similarities

In order to fully evaluate the impact of multi-grained similarities, we compare different variants of S3MA and the results are shown in Table 6. From these results, we can draw three conclusions,

- Multi-grained similarities are crucial for retrieval. Using both coarse- and fine-grained alignments in the dense space improved R@1 from 42.8 and 41.9 to 44.0 and 43.6 on text-to-video and video-to-text retrieval compared with only using coarse-grained alignment in the dense space, respectively. The same observation can be observed in the sparse space.

- Sparse space plays a crucial role in improving the alignment of modalities. We observe that incorporating coarse-grained in the dense and sparse spaces improves R@1 for text-to-video

retrieval from 42.8 to 43.3 compared to only performing coarse-grained similarity in the dense space, respectively.

- Using multi-space and multi-grained similarities simultaneously achieves the best performance. R@1 on text-to-video and video-to-text retrieval is significantly improved from 42.8 and 41.9 to 49.1 and 46.9, respectively.

### 4.3.3 Temporal Encoder and Larger Model

We also investigate the effect of the temporal encoder (TE, a small sequence transformer) and different base models. The results are shown in Table 7. S3MA with TE outperforms S3MA without TE, because it is able to better model the temporal relation among different frames in a video. Besides, using a larger base model, such as ViT-B/16, further improves the performance of S3MA, as a larger base model typically has better representation learning abilities benefiting this retrieval task as well. Similar conclusions can be found in previous works (Luo et al., 2022; Ma et al., 2022a).

### 4.4 Qualitative Results

To qualitatively validate the effectiveness of S3MA and the alignment in the sparse space, we present examples of video-to-text and text-to-video retrieval on MSR-VTT in Figures 4, 6 and 7, and the alignment in sparse space in Figure 5, respectively. The retrieval results show the satisfactory performance of S3MA, benefiting from multi-space multi-grained similarity. Notably, S3MA demonstrates precise identification of the color (*green*), objects (*bicycle*), and humans (*a man*), indicating its proficiency in capturing intricate details. In Fig-

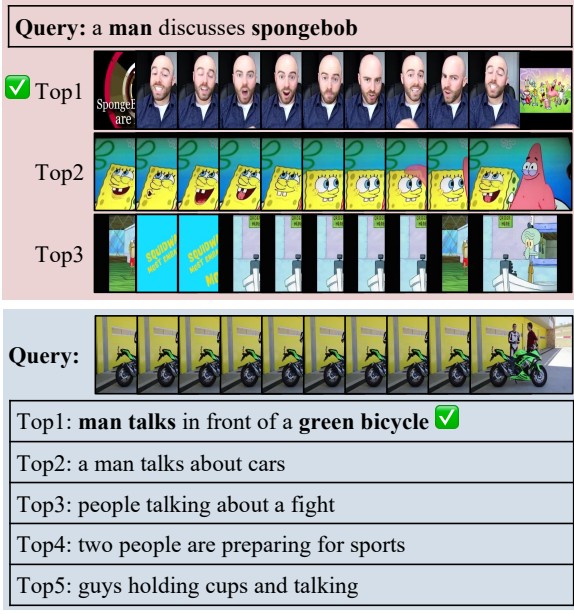

| Query: a **man** discusses **spongebob** |
|---|
| ✅ Top1 |
| Top2 |
| Top3 |

| Query: |
|---|
| Top1: **man talks** in front of a **green bicycle** ✅ |
| Top2: a man talks about cars |
| Top3: people talking about a fight |
| Top4: two people are preparing for sports |
| Top5: guys holding cups and talking |

Figure 4: Video-Text retrieval examples.

Query: a(519) movie(947) director(694) talking(248) to(1017) the(519) media(154) men(28) in(1017) press(915) conference(133) regarding(827) his(384) movie(947) and(522) hero(213) also(41)

**Video Sparse Similarity – Top10 Indices and Similarities**

| Ind | 519 | 1017 | 41 | 28 | 213 | 522 | 140 | 248 | 827 | 578 |
|---|---|---|---|---|---|---|---|---|---|---|
| Sim | 0.93 | 0.91 | 0.80 | 0.70 | 0.63 | 0.57 | 0.53 | 0.50 | 0.44 | 0.42 |

**Frame Sparse Similarity – Top10 Indices and Similarities**

| Ind | 519 | 213 | 1017 | 41 | 248 | 827 | 522 | 124 | 28 | 140 |
|---|---|---|---|---|---|---|---|---|---|---|
| Sim | 0.90 | 0.85 | 0.83 | 0.82 | 0.82 | 0.76 | 0.75 | 0.71 | 0.68 | 0.68 |

Figure 5: An example of alignment on the sparse space. The index of the concepts is shown in the brackets.

ure 5, we notice that, the video and frame features are perfectly aligned with the corresponding sparse concepts as exhibiting high similarities.

## 5 Conclusion

In this paper, to better align video and text modalities, we proposed a multi-space, multi-grained video-text retrieval framework, S3MA. Specifically, S3MA aligned different modalities in a fine-grained shared sparse space, which is initialized with a finite number of concept clusters consisting of a number of basic concepts (words) and updated in a supervised fashion with the guide of text. Besides, S3MA employed frame (fine-grained) and video (coarse-grained) features to encourage models to perform multi-grained similarity alignment. Finally, we conducted extensive experiments on three representative video-text retrieval benchmarks, showing the superiority of S3MA.

## Limitations

In the future, it would be promising to seek more fine-grained alignment, such as instance (object)-level or word-level alignment, for aligning different modalities. Moreover, our experiment focused solely on the application of sparse retrieval in video-text retrieval. It would be great to see whether sparse retrieval can help other cross-modal retrieval tasks, *e.g.*, audio-text, image-text, audio-video, and audio-image retrieval. Additionally, incorporating more detailed information such as the relationship between different objects and frames would be beneficial for the video-text retrieval problem.

Regarding the sparse space, we notice that some sparse concepts are retrieved a lot during the training procedure which might lead to the emergence of hubness (Radovanovic et al., 2010). Investigating improved clustering methods to mitigate hubness would be an interesting direction for future research. That might be due to the KNN clustering strategy and in the future and introducing better clustering strategies might be able to reduce the hubness issue, such as weighted KNN, semantic-based KNN, or part-of-speech tagging-based KNN.

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

## A Experiments

### A.1 Datasets Details

**MSR-VTT** (Xu et al., 2016) contains 10,000 videos with length varying from 10 to 32 seconds, each paired with about 20 human-labeled captions. Following the evaluation protocol from previous works Yu et al. (2018); Miech et al. (2019), we use the training-9k / test 1k-A splits for training and testing respectively.

**MSVD** (Chen and Dolan, 2011) contains 1,970 videos with a split of 1200, 100, and 670 as the train, validation, and test set, respectively. The duration of videos varies from 1 to 62 seconds. Each video is paired with 40 English captions.

**ActivityNet** (Fabian Caba Heilbron and Niebles, 2015) is consisted of 20,000 Youtube videos with 100,000 densely annotated descriptions. For a fair comparison, following the previous setting (Luo et al., 2022; Gabeur et al., 2020), we concatenate all captions together as a paragraph to perform a video-paragraph retrieval task by concatenating all the descriptions of a video. Performances are reported on the "val1" split of the ActivityNet.

### A.2 Implementation Details and Evaluation Protocols

Following Luo et al. (2022); Ma et al. (2022a), we use a standard vision transformer (Dosovitskiy et al., 2021) with 12 layers which are initialized with the public CLIP (Radford et al., 2021) checkpoints. We directly use the text encoder of CLIP as our text encoder which is also initialized with the public CLIP checkpoints.

We set the query, key, and value projection dimension size as 512 to match CLIP's output dimension and we initialize our logit scaling parameter $\lambda$ with the value from the pre-trained CLIP model. All models are optimized for 5 epochs on MSR-VTT and MSVD, and for ActivityNet, the models are trained for 20 epochs. We use AdamW (Loshchilov and Hutter, 2019) with a weight decay of 0.2 and decay the learning rate using a cosine schedule (Loshchilov and Hutter, 2017), following the method used in CLIP (Radford et al., 2021). For all experiments, we uniformly sample 12 frames from every video, resizing each frame to 224x224 as per previous works (Luo et al., 2022; Ma et al., 2022a). we set $n_{codes} = 1024$ following DiscreteCodebook (Liu et al., 2022a). To evaluate the retrieval performance of our proposed model, we use recall at Rank K (R@K, higher is better), median rank (MdR, lower is better), and mean rank (MnR, lower is better) as retrieval metrics, which are widely used in previous retrieval works (Radford et al., 2021; Luo et al., 2022; Ma et al., 2022a).

### A.3 Ablation Studies

**Evaluating the calculation of similarity between video and frame representations and cluster concepts in S3MA**. In S3MA, we use cosine similarity to calculate $sim^f$ and $sim^v$. Another way of calculating $sim^f$ and $sim^v$ might be using multi-label classification. To compare the effect of multi-label classification and cosine similarity, we conduct experiments using two multi-layer perceptrons (MLPs) with two layers and the $ReLU$ activation to predict the similarity between video and frame representations and cluster concepts. Two MLPs are also trainable. Quantitative results are shown in Table 8. Our quantitative results, shown in Table 8, indicate that the use of MLPs decreases R@1 on text-to-video and video-to-text retrieval. This suggests that cosine similarity is more suitable for VTR.

**Evaluating the importance of supervised alignment in S3MA**. In S3MA, the aligned sentence representation $\mathbf{r}_c^s$ is obtained from the text as in Eq. (1). This process aligns the sentence representation based on the instruction of the text. By doing so, the aligned sentence representation $\mathbf{r}_c^s$ can serve as the supervision (an anchor) for aligning video and frame features, providing a reference point for the alignment of different modalities. To investigate the importance of placing an anchor $\mathbf{r}_c^s$ for better alignment, we compare it to obtaining aligned sentence representation through the similarity between concept clusters $C$ and sentence feature $\mathbf{r}^t$. This alternative approach allows us to evaluate the effectiveness of using an anchor for alignment and to understand how it improves the performance of the model. To investigate the alternative approach of obtaining aligned sentence representation without an anchor, we calculate the sentence sparse space similarity $sim^t \in \mathbb{R}^{1 \times n_c}$ by calculating the cosine similarity between sentence representations and concepts as $sim_j^t = \cos(\mathbf{r}^s, C_j)$, where $sim_j^t$ is the $j$-th element of $sim^t$, $C_j$ is the $j$-th row of $C$, and $\cos$ is the cosine similarity. The aligned sentence representation $\mathbf{r}^t$ without the instruction of text is obtained by matrix multiplication as follows:

$$\mathbf{r}^t = sim^t C / \|sim^t\|_1, \qquad (4)$$

| | Text-to-Video Retrieval | | | | | Video-to-Text Retrieval | | | | |
|---|---|---|---|---|---|---|---|---|---|---|
| | R@1↑ | R@5↑ | R@10↑ | MdR↓ | MnR↓ | R@1↑ | R@5↑ | R@10↑ | MdR↓ | MnR↓ |
| S3MA (ViT-B/32) w. multi-label classification | 47.0 | 73.6 | **82.9** | 2.0 | **12.5** | 45.5 | 73.8 | **82.8** | 2.0 | **8.7** |
| S3MA (ViT-B/32) w. cosine | **49.1** | **73.9** | 82.8 | 2.0 | 13.5 | **46.9** | 73.8 | 82.1 | 2.0 | 9.3 |

Table 8: Ablation study on the calculation of similarity between video and frame representations and cluster concepts.

| | Text-to-Video Retrieval | | | | | Video-to-Text Retrieval | | | | |
|---|---|---|---|---|---|---|---|---|---|---|
| | R@1↑ | R@5↑ | R@10↑ | MdR↓ | MnR↓ | R@1↑ | R@5↑ | R@10↑ | MdR↓ | MnR↓ |
| S3MA (ViT-B/32) w/o anchor | 47.8 | 72.9 | 82.3 | 2.0 | **13.4** | 46.4 | **74.9** | 82.1 | 2.0 | **9.1** |
| S3MA (ViT-B/32) w. anchor | **49.1** | **73.9** | **82.8** | 2.0 | 13.5 | **46.9** | 73.8 | **82.1** | 2.0 | 9.3 |

Table 9: Ablation study on the instruction of text, *i.e.*, generating $\mathbf{r}_c^s$ using the similarity or the text. "w. anchor" refers to obtain $\mathbf{r}_c^s$ by text as Eq. (1). "w/o anchor" refers to obtain $\mathbf{r}_c^s$ by the similarity between sentence representations and concepts $C$ as Eq. (4)

| $\ell_{align}$ | $\ell_{alignsim}$ | Text-to-Video Retrieval | | | | | Video-to-Text Retrieval | | | | |
|---|---|---|---|---|---|---|---|---|---|---|---|
| | | R@1↑ | R@5↑ | R@10↑ | MnR↓ | MeanR↓ | R@1↑ | R@5↑ | R@10↑ | MnR↓ | MeanR↓ |
| | | 48.0 | 72.9 | 82.4 | 2.0 | 13.5 | 45.4 | 73.2 | 82.1 | 2.0 | 9.3 |
| ✓ | | 48.0 | 73.5 | 82.7 | 2.0 | **13.4** | **47.1** | **74.2** | **82.9** | 2.0 | **9.1** |
| | ✓ | 47.4 | 73.5 | 82.7 | 2.0 | 13.5 | 46.8 | 73.2 | 82.2 | 2.0 | 9.2 |
| ✓ | ✓ | **49.1** | **73.9** | **82.8** | 2.0 | 13.5 | 46.9 | 73.8 | 82.1 | 2.0 | 9.3 |

Table 10: Ablation study of $\ell_{align}$ and $\ell_{alignsim}$ on MSR-VTT based on S3MA (ViT-B/32).

| $\alpha$ | $\beta$ | Text-to-Video Retrieval | | | | | Video-to-Text Retrieval | | | | |
|---|---|---|---|---|---|---|---|---|---|---|---|
| | | R@1↑ | R@5↑ | R@10↑ | MnR↓ | MeanR↓ | R@1↑ | R@5↑ | R@10↑ | MnR↓ | MeanR↓ |
| 0.02 | 0.01 | **49.1** | 73.9 | 82.8 | 2.0 | 13.5 | **46.9** | 73.8 | 82.1 | 2.0 | 9.3 |
| 0.02 | 0.02 | 48.5 | 73.8 | **83.2** | 2.0 | 14.0 | 46.3 | 73.1 | 82.1 | 2.0 | 9.4 |
| 0.02 | 0.05 | 47.6 | 72.7 | 82.4 | 2.0 | 14.0 | 45.8 | **74.0** | 82.2 | 2.0 | 9.2 |
| 0.02 | 0.1 | 47.7 | 72.3 | 82.9 | 2.0 | 13.4 | 45.3 | 73.6 | **83.3** | 2.0 | **9.0** |
| 0.01 | 0.01 | 47.6 | 74.0 | 82.7 | 2.0 | 13.8 | 46.7 | 73.5 | 82.2 | 2.0 | 9.5 |
| 0.05 | 0.01 | 48.1 | 73.6 | 83.1 | 2.0 | **13.2** | 46.3 | 72.9 | 82.7 | 2.0 | 9.1 |
| 0.1 | 0.01 | 47.9 | **74.2** | 82.3 | 2.0 | 13.3 | 46.3 | 73.4 | 82.5 | 2.0 | 9.1 |

Table 11: Ablation study of $\alpha$ and $\beta$ on MSR-VTT based on S3MA (ViT-B/32).

| Dense Space | | | | Sparse Space | | | | Text-to-Video Retrieval | | | | | Video-to-Text Retrieval | | | | |
|---|---|---|---|---|---|---|---|---|---|---|---|---|---|---|---|---|---|
| S-V | S-F | W-V | W-F | S-V | S-F | W-V | W-F | R@1↑ | R@5↑ | R@10↑ | MdR↓ | MnR↓ | R@1↑ | R@5↑ | R@10↑ | MdR↓ | MnR↓ |
| ✓ | ✓ | | | ✓ | ✓ | | | **49.1** | **73.9** | **82.8** | 2.0 | 13.5 | **46.9** | 73.8 | **82.1** | 2.0 | **9.3** |
| ✓ | ✓ | ✓ | ✓ | ✓ | ✓ | ✓ | ✓ | 48.3 | 73.8 | 82.7 | 2.0 | **13.0** | 46.6 | **74.1** | **82.1** | 2.0 | 9.4 |
| X-CLIP | | | | | | | | 46.1 | 74.3 | 83.1 | 2.0 | 13.2 | 46.8 | 73.3 | 84.0 | 2.0 | 9.1 |

Table 12: Retrieval performance with different similarities on MSR-VTT using S3MA with the base model of ViT-B/32. "S-V", "S-F", "W-V", and "W-F" represent Sentence-Video (coarse-grained), Sentence-Frame (fine-grained), Word-Video (fine-grained), and Word-Frame (fine-grained) similarities.

where $sim^t$ is the similarity between sentence representations and concepts. The results of this comparison can be found in Table 9. The experimental results show that with the "anchor", S3MA can better align different modalities as R@1, R@5, and R@10 on text-to-video retrieval and R@1 on video-to-text retrieval have greatly improved, indicating that the supervised (anchor-based) alignment is crucial for better performance of the model.

**Effect of losses and hyperparameter sensitivity**. To further demonstrate the effectiveness of the

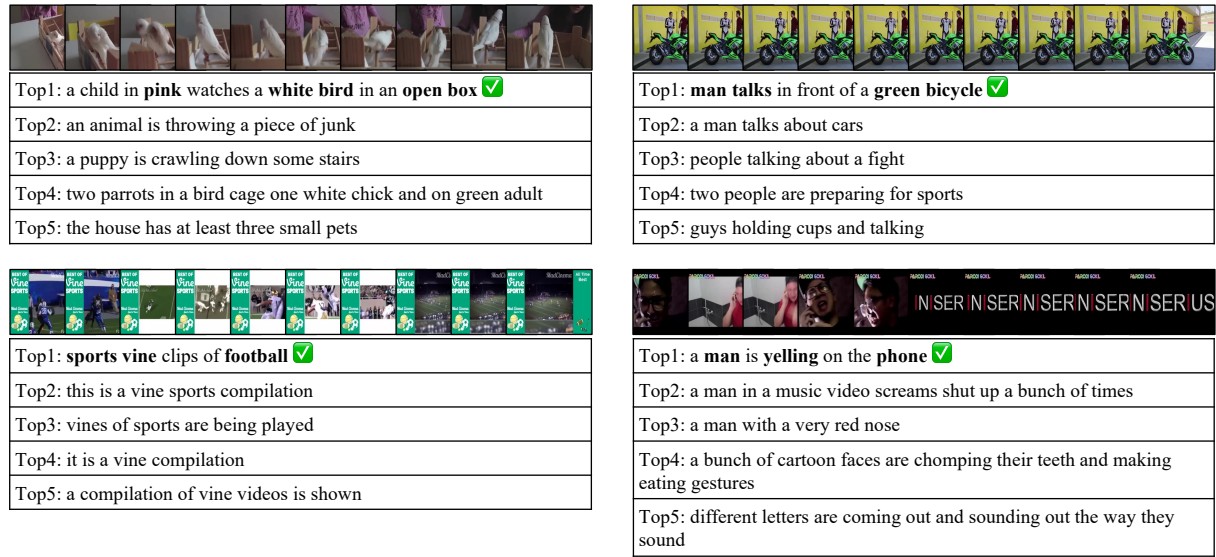

Figure 6: Top-5 video-to-text retrieval results on MSR-VTT.

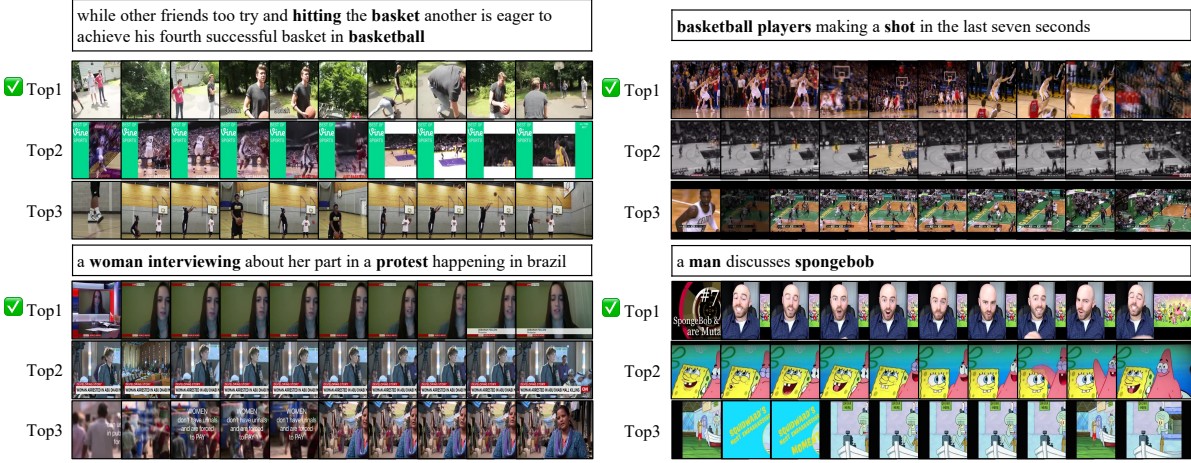

Figure 7: Top-3 text-to-video retrieval results on MSR-VTT.

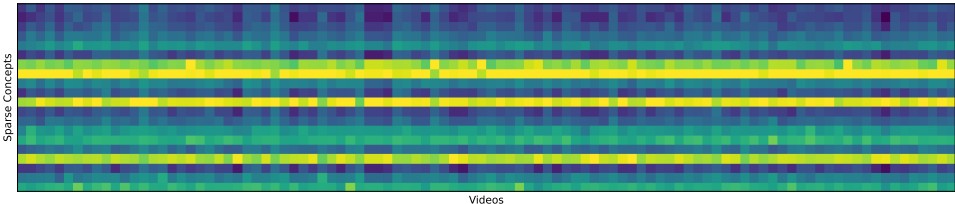

Figure 8: The activation of 20 sparse concepts by 100 randomly selected videos.

two proposed losses designed for aligning different modalities in the shared sparse space, we conduct experiments to compare the performance of these losses. The quantitative results of these experiments are shown in Table 10. The results indicate that adding both losses simultaneously achieves the best performance on the MSR-VTT dataset. When using only one loss, the performance on text-to-

video retrieval is comparable to the method without using both losses on text-to-video retrieval, but outperforms the method without the two losses on video-to-text retrieval. Specifically, when using two losses, R@1 on text-to-video retrieval and video-to-text retrieval is improved by 1.1 and 1.5, respectively. Additionally, all the other metrics, such as R@5 and R@10, are also improved, demon-

strating the power of the two proposed losses in aligning different modalities in the shared sparse space. To gain a better understanding of the sensitivity of S3MA with respect to the two hyperparameters, $\alpha$ and $\beta$, we conduct a series of experiments with different settings of $\alpha$ and $\beta$ as shown in Table 10. The results of these experiments demonstrate that, even with varying settings of $\alpha$ and $\beta$, the video-text retrieval performance remains consistent, indicating that the model is robust and not highly sensitive to these hyperparameters. This suggests that S3MA is able to achieve good performance across a wide range of settings for these hyperparameters, making it easy to adjust and optimize for specific use cases. Additionally, this also suggests that S3MA is not overly dependent on precise values of these hyperparameters, and is instead able to leverage the more important underlying features and patterns in the data.

**Are word-level features necessary?** To investigate the necessity of word-level features, we introduce word-level dense and sparse representations, along with word-frame and word-video similarities, into the dense and sparse spaces. The results are presented in Table 12. Notably, we observe a decrease in performance when incorporating word-level contrast in both dense and sparse spaces, indicating possible feature redundancy. Moreover, our approach, which incorporates word-level contrast, can be viewed as an extension of X-CLIP (Ma et al., 2022b) with the shared sparse space. We notice that contrasting representations in the aligned sparse space enhances the retrieval performance of X-CLIP.

### A.4 Aligning Examples

To show the effectiveness of S3MA, we illustrate some examples of video-to-text and text-to-video retrieval examples in Figures 4, 6 and 7. We notice that S3MA is able to align some important concepts between video and text for precise retrieval. For example, in the bottom-left video-to-text result (Figure 6), the biggest difference between the top 5 retrieved texts is "football". By precisely capturing "football" in the video, S3MA is able to give higher logits to the sentences that contain "football". Additionally, in the last (bottom-right) text to video result (Figure 7), we notice that, by understanding "man" and "discuss", S3MA is able to distinguish the top 3 retrieved videos and select the one in which a man appears. This empirically shows that

S3MA performs well in visual and textual content understanding, benefiting from multi-space and multi-grained similarity.

Moreover, we visualize the activation of sparse concepts by videos in Figure 8. We notice that, some hub sparse concepts are frequently retrieved while some are not retrieved a lot, which might be due to the KNN clustering. Moreover, we notice that the difference between activations from videos are separable.