# OpenReview forum: "Video-Text Retrieval by Supervised Sparse Multi-Grained Learning"
_EMNLP/2023/Conference — EMNLP 2023 Findings_

### Official Review · Reviewer_UH8H · 2023-08-04

**Soundness:** 1

**Excitement:**

2: Mediocre: This paper makes marginal contributions (vs non-contemporaneous work), so I would rather not see it in the conference.

**Paper Topic And Main Contributions:**

This paper proposes the shared space to alleviate the problem by the representation mismatches among the modalities (i.e., video and text).

**Questions For The Authors:**

see above

**Reasons To Accept:**

The best performances in three benchmarks in video-text retrieval
Sufficient experiments on the retrieval dataset

**Reasons To Reject:**

[-] Poor writing. To use the word 'Sparse', it is necessary to first specify the sparsity on what (e.g., quantities, distribution?). When reading the Introduction, I didn't know the sparsity of what and also didn't know why the concept of sparsity is needed or what it does.

[-] If the contributions of this paper are based on the representation enhancement of shared spaces about heterogeneous modalities, it is more convincing to validate the approach in several multi-modal video language tasks such as video question answering, video-grounded reasoning/dialogue. Why only experiments on text-video retrieval? Haven't you verified the effect on several other multi modal tasks?

[-] Does the author can visualize the effectiveness of the proposed method?

**Reproducibility:**

3: Could reproduce the results with some difficulty. The settings of parameters are underspecified or subjectively determined; the training/evaluation data are not widely available.

**Reviewer Confidence:**

3: Pretty sure, but there's a chance I missed something. Although I have a good feel for this area in general, I did not carefully check the paper's details, e.g., the math, experimental design, or novelty.

---

> ### Author Rebuttal · Authors · 2023-08-26
>
> We sincerely thank you for your time, efforts, and your detailed comments. We have carefully considered your comments and have provided responses to specific comments.
>
> **Reasons To Reject [1, What is sparse space?]:**
>
> >Your question is highly valuable, and we're grateful for the opportunity to clarify this aspect. As stated in the Abstract, ``The shared sparse space is initialized with a finite number of sparse concepts, each of which refers to a number of words`` (Line 7 to Line 9). Additionally, the definition of sparse space in our paper closely aligns with that commonly employed in text retrieval (Line 58 to Line 64 in the introduction). Recognizing the importance of conceptual clarity, we will present a more detailed and comprehensive explanation of the sparse space in our main paper, particularly within the introduction to facilitate readers' understanding of our models and the essence of our work.
>
> **Reasons To Reject [2, Why not verify the effect on other multi-modal video language tasks such as video question answering, video-grounded reasoning/dialogue.]:**
>
> >Thank you for your question. Choosing video-text retrieval is that during the inferencing, we can **directly and explicitly understand how the sparse space aids in aligning diverse modalities**, as evident in Figures 3 to 7 within our paper. These figures indicate that incorporating the sparse space does help with better aligning different modalities thus yielding better retrieval performance. As for other tasks, we have **acknowledged this potential within the limitations** section. We would like to explore the potential of the sparse space in other tasks as future work.
>
> **Reasons To Reject [3, Visualization of the effectiveness]:**
>
> >Thank you for your question, and we are delighted to direct your attention to our presented **retrieval examples showcased in Figures 3 to 7** within our main paper, appendix, and **Section 4.4 Qualitative results**. Through these visualizations, our proposed S3MA exhibits its exceptional capability in accurately identifying attributes such as color (e.g., green), objects (e.g., bicycle), and humans (e.g., a man), indicating its proficiency in capturing intricate details. In Figure 4, we notice that, the video and frame features are perfectly aligned with the corresponding sparse concepts as exhibiting high similarities.
>
> We sincerely thank you for your thoughtful comments and are open to further discussion on these matters.

---

### Official Review · Reviewer_VnYa · 2023-08-05

**Soundness:** 4

**Excitement:**

4: Strong: This paper deepens the understanding of some phenomenon or lowers the barriers to an existing research direction.

**Paper Topic And Main Contributions:**

The authors have developed an efficient, sparse method for correlating textual and visual data within a unified conceptual framework. The most important novelty of the paper, in my opinion, is to cluster the textual tokens into concepts, and align the visual and textual representations in that space.

**Questions For The Authors:**

1- How this method can handle rare and frequent concepts?
2- Is the cluster center selected as the representative?
3- Is there a way to take advantage of the cooccurrence of cluster words?
4- Why the Dense space similarity needed?

**Reasons To Accept:**

Interesting idea.
Important problem.
The solution can be applied to different types of visual-textual applications, for example phishing detection on social media, recommendation systems, and etc.
Superior results.
Well written.

**Reasons To Reject:**

KNN clustering to find concepts given words, makes the approach biased towards the word embeddings of he up-stream models. It can be claimed that the success of the networks is mostly achieved by the initial words embeddings, before clustering.

**Reproducibility:**

4: Could mostly reproduce the results, but there may be some variation because of sample variance or minor variations in their interpretation of the protocol or method.

**Reviewer Confidence:**

4: Quite sure. I tried to check the important points carefully. It's unlikely, though conceivable, that I missed something that should affect my ratings.

---

> ### Author Rebuttal · Authors · 2023-08-25
>
> We sincerely thank you for your time, efforts, and your detailed and positive comments. We have carefully considered your comments and have provided responses to specific comments.
>
> **Questions 1 [Rare and frequent concepts]:**
>
> > Thank you for your valuable questions. S3MA does not consider the frequency of words or concepts and treats both rare and frequent words or concepts equally throughout the initialization and updates. We believe that a fair or weighted strategy towards different concepts would indeed yield enhanced retrieval performance. We acknowledge the potential benefits of such an approach and have included this aspect as part of our future work. This discussion can be found in the limitations section (Line 543 to Line 553).
>
> **Questions 2 [Is the cluster center selected as the representative?]:**
>
> > Correct. During training and inferencing, the cluster centers effectively represent the words that fall within their corresponding clusters.
>
> **Questions 3 [Take advantage of the cooccurrence of cluster words?]:**
>
> > Thank you for your valuable question! We share your view that the clustering procedure could be refined by using better advanced strategies such as (cooccurrence-)weighted KNN, semantic-based KNN, or part-of-speech tagging-based KNN. We have discussed this in the section of Limitations (Line 543 to Line 553). Moreover, we are actively working on experimenting with these enhanced KNN strategies, and we will present the corresponding results in the final version of our paper.
>
> **Questions 4 [Why the Dense space similarity needed?]:**
>
> >Thank you for your valuable question! The reason why we are incorporating the dense and sparse spaces is that they excel in different aspects. The dense space (vanilla CLIP4Clip) excels in representation learning but is limited in terms of explainability. Conversely, the sparse space offers excellent explainability due to its concept-based nature, albeit with limitations in representation learning as the number of concepts are limited. And similar conclusion can be obtained from dense and sparse retrieval [1,2,3]. By integrating both spaces, we aim to leverage the best of both spaces, achieving superior retrieval performance while retaining explainability.
>
> >Moreover, we did run the experiments only using sparse space and we noticed that the Text-Video R@1 on MSR-VTT is only 38.4 which is due to the random initialization of the sparse space and the small number of learnt concepts. But it provides meaningful explanations for each concept.
> > We will include this analysis and the corresponding results in the final version of our paper.
>
> **Reasons To Reject [Success of the networks might be mostly achieved by embeddings without clustering.]:**
>
> > Thank you for your valuable comment! We reported the performance without clustering in Table 4 (Page 7) in our main paper. We notice that the **performance without clustering (48.7 on Text-to-Video R@1) is slightly lower that the performance with clustering (49.1 on Text-to-Video R@1)**. Nevertheless, clustering brings superior performance and faster inference times as the number of concepts (1024 in our experiments) is much small than the number of sub-words (approximately 30k). We agree that clustering will benefit training by acceleration and concentration of gradients while the improvements brought by clustering are limited as shown in Table 4.
>
> We sincerely thank you for your thoughtful comments and are open to further discussion on these matters.
>
> References:
>
> [1] Vladimir Karpukhin, Barlas Oguz, Sewon Min, Patrick Lewis, Ledell Wu, Sergey Edunov, Danqi Chen, and Wen-tau Yih. 2020. [Dense Passage Retrieval for Open-Domain Question Answering](https://aclanthology.org/2020.emnlp-main.550). In *Proceedings of the 2020 Conference on Empirical Methods in Natural Language Processing (EMNLP)*.
>
> [2] Kyoung-Rok Jang, Junmo Kang, Giwon Hong, Sung-Hyon Myaeng, Joohee Park, Taewon Yoon, and Heecheol Seo. 2021. [Ultra-High Dimensional Sparse Representations with Binarization for Efficient Text Retrieval](https://aclanthology.org/2021.emnlp-main.78). In *Proceedings of the 2021 Conference on Empirical Methods in Natural Language Processing (EMNLP).*
>
> [3] Yi Luan, Jacob Eisenstein, Kristina Toutanova, and Michael Collins. 2021. [Sparse, Dense, and Attentional Representations for Text Retrieval](https://aclanthology.org/2021.tacl-1.20). *Transactions of the Association for Computational Linguistics*, 9:329–345.

---

### Official Review · Reviewer_Vvmb · 2023-08-11

**Soundness:** 3

**Excitement:**

3: Ambivalent: It has merits (e.g., it reports state-of-the-art results, the idea is nice), but there are key weaknesses (e.g., it describes incremental work), and it can significantly benefit from another round of revision. However, I won't object to accepting it if my co-reviewers champion it.

**Paper Topic And Main Contributions:**

This paper introduces a multi-grained sparse learning framework designed to acquire a shared, aligned sparse space for the purpose of video-text retrieval tasks.

The authors adopt a supervised approach to learning and continuously updating the shared sparse space for text and video representations. This is achieved through the incorporation of the proposed similarity and alignment losses. Furthermore, the paper suggests incorporating a multi-grained similarity approach in the context of video retrieval tasks.

**Questions For The Authors:**

A. Table 6 showcases that the most optimal performance is attained by employing the multi-space multi-grained similarity computation. Nonetheless, it is important to underscore that the analysis regarding the influence of the introduced sparse space does not encompass its individual performance outcomes.

B. Referring to your assertion of achieving the state-of-the-art (SOTA) status, it might be appropriate to reconsider. To the best of my knowledge, several other published methods have surpassed your results, including CLIP-ViP [1], Cap4Video [2], DRL [3], and TemPVL [4].

[1] Xue, Hongwei, et al. "Clip-vip: Adapting pre-trained image-text model to video-language representation alignment."
[2] Wu, Wenhao, et al. "Cap4Video: What Can Auxiliary Captions Do for Text-Video Retrieval?"
[3] Wang, Qiang, et al. "Disentangled representation learning for text-video retrieval."
[4] Ma, Fan, et al. "Temporal perceiving video-language pre-training."

**Reasons To Accept:**

The conducted experiments aptly showcase the effectiveness of the proposed method. Notably, this paper stands out for its comprehensive ablation study and meticulous analysis of each individual module.


**Reasons To Reject:**

In essence, this paper presents an integration of global video-text and local frame-text elements within both the introduced sparse space and the original dense space, all aimed at enhancing video retrieval. However, it's worth noting that the impact of the introduced sparse space alone is not thoroughly elucidated in the analysis. The method itself should be multi-space multi-grained learning framework for video retrieval.


**Reproducibility:**

3: Could reproduce the results with some difficulty. The settings of parameters are underspecified or subjectively determined; the training/evaluation data are not widely available.

**Reviewer Confidence:**

3: Pretty sure, but there's a chance I missed something. Although I have a good feel for this area in general, I did not carefully check the paper's details, e.g., the math, experimental design, or novelty.

---

> ### Author Rebuttal · Authors · 2023-08-25
>
> We sincerely thank you for your time, efforts, and your detailed and positive comments. We have carefully considered your comments and have provided responses to specific comments.
>
> **Questions A, Reasons To Reject [Performance of only using sparse space]:**
>
> > Thank you for your valuable question! We agree that the performance comes from **multi-grained multi-space similarity** (Sec. 3.3 in the main paper). We did run the experiments only using sparse space and the Text-Video R@1 on MSR-VTT is only 38.4 which is due to the random initialization of the sparse space and by the small number of learned concepts. However, the sparse space does have a great explainability as each concept has meaning.
>
> > The dense space (vanilla CLIP4Clip) excels in representation learning but is limited in terms of explainability. Conversely, the sparse space offers excellent explainability due to its concept-based nature, albeit with limitations in representation learning as the number of concepts is limited. And similar conclusion can be obtained from dense and sparse spaces [5,6,7]. By integrating both spaces, we aim to leverage the best of both spaces, achieving superior retrieval performance while retaining explainability.
>
> > We will include this analysis and the corresponding results in the final version of our paper. Moreover, we will consider revising the title of our paper. Thank you for your suggestion again! That is valuable to us.
>
> **Questions B [Related works]:**
>
> > Thank you for your detailed comments! We will add detailed comparisons with these papers in related works and experiments in the final version of our paper. But note that, our proposed method is the **SOTA without any auxiliary data, models, or post-processing methods**. Our S3MA achieves **51.7 R@1 on Text-to-Video (ViT-B/32)** on MSR-VTT with DSL [8]. We will carefully revise our paper to avoid any confusion and add related experimental results in our final version of the paper.
>
> > We acknowledge the improved performance achieved by models [1,2,3] leveraging auxiliary data, models, or post-processing methods achieves better retrieval performance, while TemPVL[4] is a pre-training task with lower retrieval performance on MSR-VTT. However, we will revise our paper and **carefully compare with these papers [1,2,3,4] for fair comparison** in our final version of the paper.
>
> We sincerely thank you for your thoughtful comments and are open to further discussion on these matters.
>
> [1] Xue, Hongwei, et al. "Clip-vip: Adapting pre-trained image-text model to -language representation alignment."
>
> [2] Wu, Wenhao, et al. "Cap4: What Can Auxiliary Captions Do for Text- ?"
>
> [3] Wang, Qiang, et al. "Disentangled representation learning for text- ."
>
> [4] Ma, Fan, et al. "Temporal perceiving -language pre-training.”
>
> [5] Vladimir Karpukhin, Barlas Oguz, Sewon Min, Patrick Lewis, Ledell Wu, Sergey Edunov, Danqi Chen, and Wen-tau Yih. 2020. [Dense Passage Retrieval for Open-Domain Question Answering](https://aclanthology.org/2020.emnlp-main.550). In *Proceedings of the 2020 Conference on Empirical Methods in Natural Language Processing (EMNLP)*.
>
> [6] Kyoung-Rok Jang, Junmo Kang, Giwon Hong, Sung-Hyon Myaeng, Joohee Park, Taewon Yoon, and Heecheol Seo. 2021. [Ultra-High Dimensional Sparse Representations with Binarization for Efficient Text ](https://aclanthology.org/2021.emnlp-main.78). In *Proceedings of the 2021 Conference on Empirical Methods in Natural Language Processing (EMNLP).*
>
> [7] Yi Luan, Jacob Eisenstein, Kristina Toutanova, and Michael Collins. 2021. [Sparse, Dense, and Attentional Representations for Text ](https://aclanthology.org/2021.tacl-1.20). *Transactions of the Association for Computational Linguistics*, 9:329–345.
>
> [8] Xing Cheng, Hezheng Lin, Xiangyu Wu, Fan Yang, and Dong Shen. 2021. Improving video-text retrieval by multi-stream corpus alignment and dual softmax loss. CoRR, abs/2109.04290.

---

### Meta-Review · Area_Chair_2m2u · 2023-09-22

**Recommendation:** 3

**Metareview:**

This aper presents a framework in which sparsification is employed to efficiently correlate and align textual and visual data for improved video-text retrieval. The most novel contribution of the paper is to cluster the textual tokens into concepts, and align the visual and textual representations in that space. Reviewers agree this presents an interesting idea, and is being applied to an important problem, while being applicable to further types of visual-textual applications, for example social media, recommendation systems, and etc. Authors achieve good results and discuss relationship with related work in-depth during the rebuttal period. Overall the paper is well written, and would be further strengthened by the incorporation of the feedback and the discussion.

One reviewer mentions that a non-anonymized version of the paper has been published *before* the anonymization period, although I was not able to find it - but this would be fine in any case.
While reviewers are split in their evaluation, the AC agrees with the reviewers that explicitly state that they find this paper "worthy of acceptance" and that evaluation on additional datasets/ tasks does not seem warranted. Given the strength of the results, the AC's recommendation is for acceptance to Findings, while acceptance to the main conference would require stronger results (either better on-task performance, or other clear advantages of the approach such as efficiency).

---

### Decision · Program_Chairs · 2023-10-07

**Decision:**

Accept-Findings

**Comment:**

This aper presents a framework in which sparsification is employed to efficiently correlate and align textual and visual data for improved video-text retrieval. The most novel contribution of the paper is to cluster the textual tokens into concepts, and align the visual and textual representations in that space. Reviewers agree this presents an interesting idea, and is being applied to an important problem, while being applicable to further types of visual-textual applications, for example social media, recommendation systems, and etc. Authors achieve good results and discuss relationship with related work in-depth during the rebuttal period. Overall the paper is well written, and would be further strengthened by the incorporation of the feedback and the discussion.

One reviewer mentions that a non-anonymized version of the paper has been published *before* the anonymization period, although I was not able to find it - but this would be fine in any case.
While reviewers are split in their evaluation, the AC agrees with the reviewers that explicitly state that they find this paper "worthy of acceptance" and that evaluation on additional datasets/ tasks does not seem warranted. Given the strength of the results, the AC's recommendation is for acceptance to Findings, while acceptance to the main conference would require stronger results (either better on-task performance, or other clear advantages of the approach such as efficiency).